# Exploring Host-Specificity: Untangling the Relationship between *Leishmania* (*Viannia*) Species and Its Endosymbiont *Leishmania RNA Virus* 1

**DOI:** 10.3390/microorganisms11092295

**Published:** 2023-09-12

**Authors:** Mayara Cristhine de Oliveira Santana, Khaled Chourabi, Lilian Motta Cantanhêde, Elisa Cupolillo

**Affiliations:** 1Leishmaniasis Research Laboratory, Oswaldo Cruz Institute, Oswaldo Cruz Foundation, Rio de Janeiro 21040360, Brazil; mayara.santana@ioc.fiocruz.br (M.C.d.O.S.); lilian.cantanhede@ioc.fiocruz.br (L.M.C.); 2Instituto Nacional de Ciência e Tecnologia de Epidemiologia da Amazônia Ocidental, INCT EpiAmO, Porto Velho 76812100, Brazil

**Keywords:** *Leishmania* (*Viannia*), *Leishmania RNA Virus* 1, phylogeny, host specificity, coevolution

## Abstract

A relevant aspect in the epidemiology of Tegumentary Leishmaniasis (TL) are the *Leishmania* parasites carrying a viral endosymbiont, *Leishmania RNA Virus* 1 (LRV1), a dsRNA virus. *Leishmania* parasites carrying LRV1 are prone to causing more severe TL symptoms, increasing the likelihood of unfavorable clinical outcomes. LRV1 has been observed in the cultured strains of five *L.* (*Viannia*) species, and host specificity was suggested when studying the LRV1 from *L. braziliensis* and *L. guyanensis* strains. The coevolution hypothesis of LRV1 and *Leishmania* was based on phylogenetic analyses, implying an association between LRV1 genotypes, *Leishmania* species, and their geographic origins. This study aimed to investigate LRV1 specificity relative to *Leishmania* (*Viannia*) species hosts by analyzing LRV1 from *L.* (*Viannia*) species. To this end, LRV1 was screened in *L.* (*Viannia*) species other than *L. braziliensis* or *L. guyanensis*, and it was detected in 11 out of 15 *L. naiffi* and two out of four *L. shawi*. Phylogenetic analyses based on partial LRV1 genomic sequencing supported the hypothesis of host specificity, as LRV1 clustered according to their respective *Leishmania* species’ hosts. These findings underscore the importance of investigating *Leishmania* and LRV1 coevolution and its impact on *Leishmania* (*Viannia*) species dispersion and pathogenesis in the American Continent.

## 1. Introduction

Endosymbiotic viruses are found in different *Leishmania* species. Although some *Leishmania* species present an association with viruses from the families *Leishbunyaviridae* [1] and *Narnaviridae* [2], several *Leishmania* species human pathogens bear viruses from the *Totiviridae* family, and these are named *Leishmaniavirus*; it is a double-strand RNA virus classified into two species, *Leishmaniavirus* 1 (LRV1) and *Leishmaniavirus* 2 (LRV2), and the two species are associated with *Leishmania* (*Viannia*) and *Leishmania* (*Leishmania*) species, respectively [3]. Similar genetic relationships observed for *Leishmania* species are observed for LRV1 and LRV2 [4].

Cutaneous leishmaniasis (CL) is a neglected disease that primarily affects impoverished populations in developing countries [5]. In the American Continent, Brazil holds the highest incidence of CL. The disease involves several *Leishmania* species: *L.* (*Leishmania*) *amazonensis* and others within the subgenus *Viannia*—*L.* (*Viannia*) *guyanensis*, *L.* (*V.*) *braziliensis*, *L.* (*V.*) *shawi*, *L.* (*V.*) *lainsoni*, *L.* (*V.*) *naiffi*, and *L.* (*V.*) *lindenbergi* [6]. A significant aspect of CL epidemiology in Brazil, and possibly beyond, is the presence of LRV1 [7,8]. This viral endosymbiont may influence the outcome and severity of CL [8,9,10,11,12,13,14]. Previously, in an experimental model, it was demonstrated that *Leishmania* infections harboring high burdens of LRV1 exhibit a metastatic infection profile. This is instigated by the recognition of the virus via the host’s Toll-like Receptor 3 pathway, coupled with the ability to undermine the host’s immune response via the same pathway. Consequently, this leads to the sustained presence of LRV1+ parasites in cutaneous lesions [13].

LRV1 is associated with exacerbating infections, driving the mucosal form of leishmaniasis. This viral role relies on the production of Type I interferons (Type I IFNs) by macrophages [15,16]. LRV1 subverts innate immunity, negatively regulating the NLRP3 inflammasome and thereby promoting parasite survival and chronic disease [16]. Notably, the presence of LRV1 increases the risk of mucosal development by nearly threefold compared to CL cases without the virus [8]. However, while some studies establish a link between LRV1 and mucosal manifestations, others show associations with therapeutic failure [17,18,19]. Differences in the parasites circulating in the different regions were evaluated and/or in the LRV1 infecting these parasites, which are possible explanations for the observed differences among some studies.

Previous studies have highlighted host-specific interactions between LRV and *Leishmania* species. A monophyletic group was observed for LRV2 from *L. major* and *L. aethiopica*, raising the idea of the co-evolution of LRV2 and *Leishmania* (*Leishmania*) species [20]. The co-evolution of LRV1 and *L.* (*Viannia*) species was also proposed based on evidence of host specificity between *L.* (*Viannia*) species and LRV1 genotypes [21].

The genomic sequences of LRV1 from various *Leishmania* species suggest specificity in *L.* (*Viannia*) species–LRV1 interactions [21,22]. For instance, LRV1 from *L.* (*V.*) *shawi* was akin to LRV1 from *L. guyanensis*, corroborating the similarity observed between these two *Leishmania* species [21,23]. Although LRV1 was detected already in the cultivated strains of *L. naiffi* and *L. panamensis* [18,24,25,26], there is no information on the phylogenetic relationship of viruses from these species compared to LRV1 from other species. LRV1 was already associated with human infections caused by *L. lainsoni* [8] and *L. peruviana* [27], but not in the cultivated strains from these species, thus limiting the comparisons with LRV1 from other species.

Given the importance of *Leishmania*-LRV1 symbiosis in the epidemiology of cutaneous and mucosal leishmaniasis, a comprehensive understanding of the virus’s diversity and spread within parasite populations is crucial. Thus, our study aimed to examine the presence of LRV1 in various strains of Brazilian *L.* (*Viannia*) species, excluding *L. braziliensis* and *L. guyanensis*, since many LRV1 sequences are available for these two species. To enhance our understanding of LRV1’s interactions across *Leishmania* species, a comparative analysis was performed between the available sequences of *L. braziliensis* and *L. guyanensis*, along with newly acquired LRV1 data from *L. naiffi* and *L. shawi*.

## 2. Materials and Methods

### 2.1. Leishmania Culture

Strains from *L. naiffi* (*n* = 18), *L. lainsoni* (*n* = 4), *L. shawi* (*n* = 4), *L. lindenbergi* (*n* = 3), and *L. utingensis* (*n* = 1)—from different geographic regions and available at the *Leishmania* Collection of Fiocruz (CLIOC)—were screened (Appendix A). Parasites were grown in NNN (Novy–MacNeal–Nicolle), and Schneider medium supplemented with 20% fetal bovine serum and incubated in a BOD (biochemical oxygen demand) incubator at 25 °C until reaching the average amount of 5 × 10^6^ parasites.

Cultures were centrifuged at 1400× *g* for 10 min at 4 °C, resuspended in DNA/RNA Shield™ (Zymo Research Corporation—Irvine, CA, USA), and stored at −20 °C until RNA extraction.

### 2.2. RNA Extraction and cDNA Synthesis

The RNA of the strains was extracted using the TRIzol^®^ reagent (Invitrogen—Carlsbad, CA, USA). RNA concentration and purity were determined using a NanoDrop^®^ 2000 spectrophotometer (Thermo Scientific™—Wilmington, CA, USA). Reverse transcription was performed using 2 μg of RNA and the High-Capacity cDNA Reverse Transcription Kit (Applied Biosystems™—Foster City, CA, USA) following the manufacturer’s recommendations.

### 2.3. LRV1 Detection

An LRV1 screening protocol was performed using primers LRV F–5′-ATGCCTAAGAGTTTGGATTCG-3′ and LRV R–5′-ACAACCAGACGATTGCTGTG-3′ [8] (Figure 1). *L. guyanensis* (MHOM/BR/1975/M4147) and *L. braziliensis* (MHOM/BR/1975/M2903) strains were used as positive and negative controls for the experiments, respectively [28,29]. For *Leishmania* HSP70 fragment amplification, which was used as an endogenous control of all RT-PCR reactions, primers Hsp70cF 5-GGACGAGATCGAGCGCATGGT-3′ and Hsp70cR 5′-TCCTTCGACGCCTCCTGGTTG-3′ were used [30]. For both reactions, a final volume of 50 μL was used: 10X Buffer (1X), MgCl_2_ (1.5 mM), dNTPs (0.2 mM), Primer F (0.2 μM), Primer R (0.2 mM), Taq Platinum (1.0 U/μL), and 1 μL of cDNA. RT-PCR was performed at 94 °C for 2 min, 35 cycles at 94 °C for 30 s, 58 °C for 30 s, and 72 °C for 45 s, with a final extension phase at 72 °C for 5 min. 

Positive strains for LRV1 (*n* = 12; 11 *L. naiffi* and 1 *L. shawi*) were submitted to another PCR reaction that aimed to amplify a fragment of approximately 850 base pairs for LRV1, which is a phylogenetically informative fragment. For this, primers LRV1 F *orf1* 5′-ATGCCTAAGAGTTTGGATTCG-3′ and LRV1 R *orf2* 5′-AATCAATTTTCCCAGTCATGC-3′ [21] were used, amplifying a fragment corresponding to a part of the *orf1* region and the beginning of the *orf2* region, including the portion responsible for encoding the viral capsid protein (Figure 1). A final volume of 50 μL was used: 10X Buffer (1X), MgCl_2_ (1.5 mM), dNTPs (0.2 mM), Primer F (0.2 μM), Primer Rg (0.2 mM), Taq Platinum (1.0 U/µL), and 3 µL of cDNA. RT-PCR was performed at 95 °C for 2 min, followed by 35 cycles at 95 °C for 30 s, 57 °C for 45 s, and 72 °C for 45 s, with a final extension phase at 72 °C for 5 min. All RT-PCR products were stained using GelRed^®^ (Biotium—Fremont, CA, USA) and visualized in a 2% agarose gel.

### 2.4. Sequencing

For sequencing, 45 μL of RT-PCR products (850 bp) was purified using the Wizard^®^ SV Gel and RT-PCR Clean-Up System kit (Promega—Madison, WI, USA) following the manufacturer’s recommendations. Sanger sequencing was performed on the Fiocruz DNA Sequencing Platform—Rio de Janeiro (RPT01A).

### 2.5. Analyses of LRV1 Sequences

The consensus sequences were created using the BioEdit program [31]. Then, the inference of the best model to build the phylogenetic tree was performed in MEGA X [32] software, where the Tamura 3-parameter model with gamma distribution parameter (G) and invariable sites (I) presented the lowest BIC (Bayesian information criterion) score. The tree was constructed using the maximum likelihood method, employing 10,000 replicates (bootstrap). 

In addition, to increase the chances of finding the most parsimonious connections, networks using NeighborNet were built using the SplitsTree program [33]. For that, we used the MEGAX function to exclude sites with missing/ambiguous data and those with sequence gaps, which resulted in sequences that were 509-nucleotide-long and were common across the groups of each species.

The analyzed sequences correspond to the LRV1 detected in *L. naiffi* and *L. shawi* strains that were isolated from sandflies and humans from Amazonian regions, and these were already deposited in GenBank (Table 1). The analysis was conducted by comparing the new LRV1 sequences to those that are already available (Appendix A). 

## 3. Results

### 3.1. LRV1 Was Not Detected in All L. (Viannia) Species Analyzed but Was Frequent in L. naiffi Strains

All RT-PCR reactions were performed with the promastigotes of available strains identified as *L. naiffi* (*n* = 18), *L*. *shawi* (*n* = 4), *L. lainsoni* (*n* = 4), *L. lindenbergi* (*n* = 3), and *L. utingensis* (*n* = 1). Of these, seven *L. naiffi*, three *L*. *shawi*, and all *L. lainsoni*, *L. lindenbergi* and *L. utingensis* strains were negative for the virus. LRV1 was detected in 11 *L. naiffi* strains (61%) and one analyzed *L*. *shawi*. The geographical distributions of all analyzed strains that are negative or positive for LRV1 are demonstrated in Figure 2. After quality checking, sequences ranging from 632 nt to 799 nt (Table 1) were employed in the analyses described below.

### 3.2. Variability of LRV1 Diversity across Leishmania Host Species

In addition to the sequences obtained in the present study, another 47 LRV1 sequences from *L*. *guyanensis* (*n* = 35), *L*. *braziliensis* (*n* = 11), and *L*. *shawi* (*n* = 1) strains available on GenBank were included in the analyses (Appendix A), corresponding to the sequences reported from studies in French Guiana [22], Bolivia [34,35], and Brazil [21,33,35]. Considering the LRV1 sequences analyzed, the same level of diversity was observed within LRV1 from *L*. *guyanensis* and LRV1 from *L*. *braziliensis*, and LRV1 sequences from *L*. *naiffi* were highly similar, such as those for *L*. *shawi* (Table 2). The level of similarity observed among sequences from *L. guyanensis* clustering in Groups A, B, D, and E was similar to those observed for *L. naiffi* and *L. shawi*. However, the similarities for LRV1 sequences from *L. braziliensis* were lower and similar to those observed for *L. guyanensis* when considering Groups A, B, and C as one group, based on the phylogenetic tree (Appendix A; Figure 3).
Figure 3Maximum likelihood phylogenetic tree of *Leishmania RNA Virus* 1 found in *Leishmania* (*Viannia*) species. The boxes detail the new LRV1 sequences presented in this study that are from *L. shawi* (in purple) and *L. naiffi* (in blue). The evolutionary history was inferred by using the maximum likelihood method and the Tamura 3-parameter model [36]. The tree with the highest log likelihood (−4303.97) is shown. The percentage of trees in which the associated taxa clustered together is shown next to the branches. Initial tree(s) for the heuristic search were obtained automatically by applying Neighbor-Join and BioNJ algorithms to a matrix of pairwise distances that were estimated using the Tamura 3-parameter model and then selecting the topology with the superior log likelihood value. A discrete gamma distribution was used to model evolutionary rate differences among sites (5 categories (+*G*, parameter = 0.8242). The rate variation model allowed for some sites to be evolutionarily invariable (+*I*, 28.96% sites). The tree is drawn to scale, with branch lengths measured in terms of the number of substitutions per site. This analysis involved 59 nucleotide sequences. There were a total of 442 positions in the final dataset. Ls = *L*. *shawi*; Ln = *L. naiffi*. For details of each strain, see Table 1 and Appendix A. * Groups are defined following the proposal of Tirera et al. [22]. Groups G and H are defined in the present study.
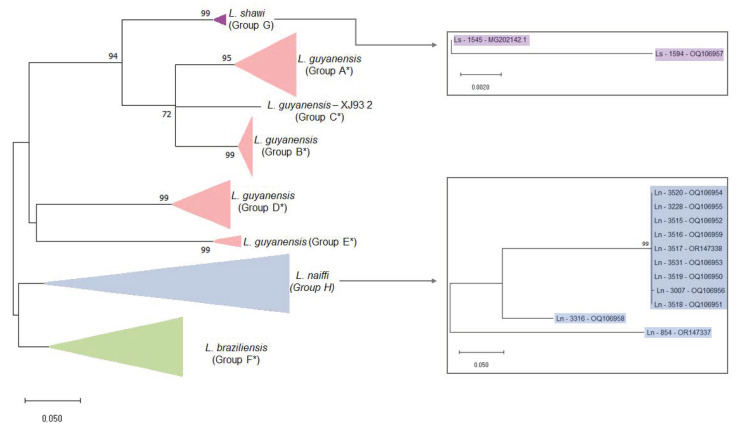



### 3.3. Higher Similarity Is Observed for LRV1 Sequences among Closely Related Leishmania Species

Putting together the results obtained in this study and the sequences of LRV1 that are publicly available, it was possible to compare LRV1 from four different species: *L*. *guyanensis*, *L*. *braziliensis*, *L*. *naiffi*, and *L*. *shawi*. The analyses between the group mean distance reveals that *L*. *shawi* and *L*. *guyanensis* exhibit lower distances, particularly within Groups A, B, and C (Appendix A). However, there is considerable diversity among LRV1 from *L*. *guyanensis*, with *L*. *braziliensis* and *L*. *naiffi* showing a significant distance between them (Table 3). 

### 3.4. Host-Specificity Is Clearly Observed in the LRV1-L. (Viannia) Species Relationship

To investigate phylogenetic relationships between LRV1 from *L. naiffi* and *L. shawi*, along with the other 47 LRV1 sequences available (from *L. guyanensis* and *L. braziliensis* and only one *L. shawi*), a maximum likelihood (ML)-based tree (Figure 3) and phylogenetic network (Figure 4) were constructed. The maximum likelihood tree shows several groups with strong support (higher than 70%), and the results are in agreement with previous studies [21,22], indicating that LRV1 from the *L. guyanensis* grouping in a cluster comprised different subclusters (named accordingly to Tirera et al. [22] here) and only one cluster for LRV1 was observed to be from *L. braziliensis*, despite the variability observed within this group (Figure 3 and Table 2). The new LRV1 sequence obtained for *L. shawi* clustered together with the other sequence available, keeping a close relationship with the *L. guyanensis* cluster (Figure 3). Although a clear group is observed for LRV1 sequences from *L. naiffi*, two strains presented very divergent LRV1 sequences and were grouped with other LRV1s from *L. naiffi* with low bootstrap support, but the most parsimonious connections for these two sequences were observed with the LRV1 from *L. naiffi* (Figure 4). These two strains are from Pará, and all the others are from another but the same endemic region. Although there is a clear clustering of LRV1 sequences based on their *Leishmania* host species, there is no association with the geographic distribution of the analyzed sptrain. For example, strains from cluster D (LRV1 from *L. guyanensis*) were observed in Manaus, Rondônia, and Pará, and strains from cluster F (LRV1 from *L. braziliensis*) were observed in Bolivia, Rondônia, and French Guyana (Figure 2).

The phylogenetic network shows sequences of clustering similar to those observed with the ML tree. The network, however, suggests a common ancestor relative to the structure for all LRV1s (Figure 4). The structure could be the result of the low amount of recombination between the sequences of the network, representing a bottleneck, or it may be simply a result of divergent phylogenetic signals. 

## 4. Discussion

Virus-like particles were demonstrated in *Leishmania* parasites in the late 1970s [37] but the first molecular description of *Leishmania RNA Virus* was in 1988 for viruses found in the cytoplasm of an *L*. (*V*.) *guyanensis* strain [28]. Ever since LRV1 was detected in clinical isolates from Peru [38], Brazil [8,18], Colombia [7], Bolivia [19], Costa Rica [39], French Guiana [40], and more recently in Panamá [24], these viruses were observed not only in *L*. *guyanensis* but also in strains identified as *L*. *braziliensis*, *L*. *shawi*, *L. naiffi*, and *L*. *panamensis* [8,21,24,25,26,41,42], indicating an old relationship between LRV1 and the *L*. (*Viannia*) subgenus. Furthermore, LRV1 was detected in clinical samples collected from patients infected by *L*. *lainsoni* [8] and *L*. *peruviana* [27], but no cultivated strains of these two strains that are positive for LRV1 are available yet, limiting our analyses. Considering the epidemiological and medical importance of the symbiosis between *Leishmania* and LRV, there is a distinct requirement to comprehend the variety and spread of the virus within parasite populations.

The coevolution hypothesis for the LRV-*Leishmania* species emerged in 1995 when Widmer and Dooley carried out a phylogenetic analysis and found that the genetic distances between LRV types that mirrored the heterogeneity observed for *Leishmania* species based on random amplified polymorphic DNA (RAPD) fingerprints [4]. More than ten years later, a study presenting a genetic characterization and phylogenetic analysis of LRV1 sequences from 27 *L*. *guyanensis* strains and two *L*. *braziliensis* was published, and host specificity for LRV1 began to be revealed [22]. A year later, a robust phylogenetic analysis was presented, including 35 LRV1 sequences from *L*. *guyanensis*, 11 from *L*. *braziliensis*, and for the first time, a sequence of LRV1 was found in an *L*. *shawi* strain [21]. Both studies presented evidence corroborating the hypothesis of the coevolution of LRV1 and *L.* (*Viannia*) parasites, grouping LRV1 sequences according to their host–parasite species. Host specificity was also demonstrated for LRV2-infecting *L. major* and *L. aethiopica* [20].

The abovementioned studies, which suggest a specific relationship between LRV1 and *L*. (*Viannia*) species, combined with the observation of LRV1 in other species, motivate the present study in screening for LRV1 in strains representing *L*. (*Viannia*) species that have not been analyzed so far and that are available in the *Leishmania* Collection from Fiocruz. To this end, we analyzed all available strains for *L*. *lainsoni*, *L*. *lindenbergi*, *L. naiffi*, *L*. *shawi*, and *L*. *utingensis*. As previously demonstrated, LRV1 was detected in *L. naiffi* and *L*. *shawi* strains [21,43] but not in *L*. *lainsoni*, despite the fact that LRV1 was previously detected in clinical samples that were collected from a patient presenting cutaneous leishmaniasis caused by this species [8]. Of note, *Leishmania* parasites were isolated from this patient, and the identified strain was included in our analyses (IOCL 3398), but it was negative for LRV1. There are some possibilities for explaining these results, including the possibility of mixed infection due to two or more *Leishmania* species but with isolation and growth in the culture medium of *L*. *lainsoni* to the detriment of another *L*. (*Viannia*) species that does not grow very well in a culture medium such as *L*. *lainsoni* [44]. The loss of LRV1 during the process of cultivation is also another possibility [45]. Thus, herein, we were able to screen LRV1s in different strains from different *L*. (*Viannia*) species, and we were able to analyze the nucleotide sequences of LRV1 from *L. naiffi* and *L*. *shawi*. Interestingly, more than 50% of analyzed *L. naiffi* strains were positive for LRV1, but we do not know yet if this symbiotic relationship confers any advances to *L. naiffi* parasites, such as the capacity of interacting with different sandfly species and/or dispersion in different geographic regions [46]. It was demonstrated that a *L. naiffi* infection cannot have a self-healing nature, as described years ago [47,48]. Patients could experience a poor response to antimonial or pentamidine therapy [43]. A case of a patient who was infected by *L. naiffi* carrying LRV1 was only first reported in 2019, raising the possibility that the presence of this virus could increase *Leishmania* spp. virulence and thereby influence therapeutic failure [18], which are aspects already observed for *L*. *braziliensis* and *L*. *guyanensis* but need further investigation. Similarly, the first *L. shawi* infection in mucosal secretion was recently observed in Brazil, and it represents a warning for the possible association between *L. (V.) shawi* and mucosal lesions [49]. 

Although it is still important to investigate the specificity of LRV1 relative to other *Leishmania* species, such as *L*. *panamensis*, since LRV1 is already detected in cultivated strains of this species [24,25,26], our results strongly support this kind of relationship, keeping *L*. *shawi* in a separate cluster that is closely related to *L*. *guyanensis*. Here, we assumed the groups suggested by Tirera et al. 2017 [22], where *L. guyanensis* was divided into five subclusters (A–E). The divergence within *L*. *guyanensis* is higher than that observed between *L*. *guyanensis* and *L*. *shawi*, corroborating the assumption that *L*. *guyanensis* comprises complex species [23]. Following this, LRV1 sequences from *L. shawi* formed another subcluster, named here as Group G, and this subcluster is closely related to the *L. guyanensis* subclusters A, B, and C. Although LRV1 sequences were obtained for only two *L. shawi* strains so far, the fact that they clustered together despite the strains being from different geographic regions is also an important aspect that supports host specificity for LRV1. The diversity of LRV1 from *L. guyanensis*, which forms several subclusters, must be better explored, but the number of LRV1 sequences from the analyzed *L*. *guyanensis* and the geographic dispersion of these parasites in the Amazon region might contribute to this observation.

Host specificity was also observed for LRV1 genotypes from *L. naiffi*. The phylogenetic tree and NeighborNet (Figure 3 and Figure 4) show that LRV1 sequences from *L. naiffi* were clustered in a well-supported monophyletic clade. Of note, most LRV1 sequences from the analyzed *L. naiffi* were very similar; in contrast, few differences were observed between the two LRV1 sequences from the analyzed *L*. *shawi*, despite being isolated from different hosts in different regions. The diversity observed within LRV1 sequences from the same species must be further investigated, but it is important to consider that all but one *L. naiffi* strain, presenting highly similar LRV1s, were obtained from patients who were infected in the same endemic region and who were included in the same study, suggesting possible problems during laboratory manipulation. However, this very similar group contained one strain that was previously isolated before the mentioned study, and it was not manipulated together with the other strains, suggesting a homogeneity for the *L. naiffi* population circulating within this area and causing human disease, which can represent an epidemic clone. The high similarity among LRV1 sequences from *L*. *naiffi* is also interesting, since only LRV1 sequences obtained for two *L. naiffi* strains, IOCL 3316 and IOCL 854, which are both isolated in Pará, showed a different phylogenetic pattern compared with all other *L. naiffi* strains (*n* = 9) isolated from Manaus (Amazonas—Brazil). The IOCL 854 strain was obtained from a sandfly species, *Lutzomyia squamiventris*, and the LRV1 from this strain reflected a basal position relative to the *L. naiffi* clade, despite the close relationship of this strain to other *L. naiffi*, including IOCL 3007, as previously demonstrated [23,50]. Considering the *Leishmania* (*Viannia*) species was depicted by microsatellite analyses, it is expected that parasites from populations that circulate the Amazon basin (POP1 and POP3 after Kuhls et al. [50]) carry LRV1, and each subpopulation has an association with specific LRV1 genotypes. 

By analyzing LRV1 sequences from several strains that represent different *L. (Viannia)* species, we demonstrated LRV1 genotypes form distinct clusters corresponding to their *Leishmania* species’ host, suggesting that the transfer of viral particles between strains from different species does not occur frequently. Altogether, our results reinforce the concordance between the phylogenetic patterns of LRV1 and *Leishmania (Viannia)* species, providing support for the prevailing hypothesis that LRV1 is an ancient virus that has undergone co-evolution with their hosts [3,4]. Recently, it was shown that parasite hybridization might explain the high occurrence of the symbiotic interaction between LRV1 and *L. braziliensis* in Peru and Bolivia [51]. It is possible that this also explains the high frequency of LRV1 in parasites from the Brazilian Amazon Region, since many possible hybrids were described in the region [52], and analyses of microsatellite markers have shown extensive diversity in the subgenus *L. (Viannia)*, with an indication of both clonality and recombination as a strategy of reproduction [50].

## 5. Conclusions

Our study adds to the growing body of evidence supporting a specific relationship between *Leishmania RNA Virus* 1 (LRV1) and species within the *L*. (*Viannia*) subgenus. LRV1 was detected in various clinical isolates from different species, including *L*. *guyanensis*, *L*. *braziliensis*, *L*. *shawi*, *L. naiffi*, and *L*. *panamensis*, but not in other species and not in the *L*. *braziliensis* circulating outside the Amazon Basin, raising intriguing questions about host specificity and the potential impact on virulence and therapeutic response. 

While the results provide strong support for the association between LRV1 and specific *Leishmania* species, more investigations are needed to understand its specificity relative to other species, such as *L*. *panamensis*. The divergence observed within LRV1 sequences from the same species warrants further scrutiny, especially considering potential issues during laboratory manipulation and the homogeneity of *L. naiffi* populations in certain endemic regions. Overall, the identification and characterization of LRV1 in different *Leishmania* species shed light on the complex interactions between these viruses and the parasites that they infect. Future research in this area may uncover novel insights into the biology and pathogenesis of *Leishmania* infections, offering new perspectives on therapeutic strategies and disease management.

## Figures and Tables

**Figure 1 microorganisms-11-02295-f001:**
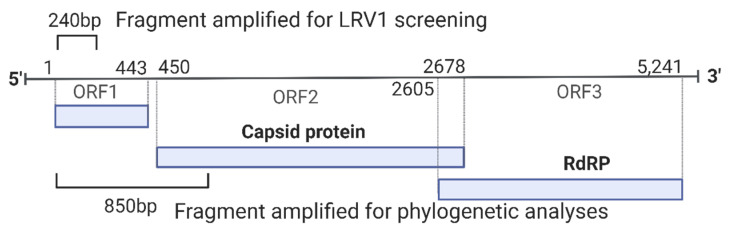
Schematic representation of the genome of LRV1 showing the region amplified for each primer used for screening (240 bp) and phylogenetic analyses (850 bp). RdRP: RNA-dependent RNA polymerase. Genome length based on NCBI RefSeq NC_003601.

**Figure 2 microorganisms-11-02295-f002:**
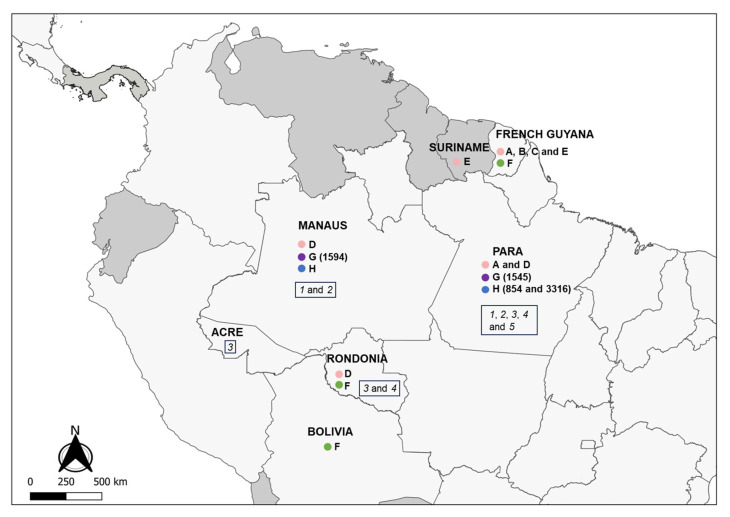
Partial map of South and Central America indicating the countries and Brazilian states where LRV1 has been detected in the *Leishmania* (*Viannia*) spp. analyzed in this study. Colored circles refer to species presenting LRV1: pink = *L*. *guyanensis*; green = *L*. *braziliensis*; purple = *L*. *shawi*; blue = *L*. *naiffi*. Letters (A to H) indicate groups that are defined according to the phylogenetic analyses (Figure 3); numbers indicate the IOCL for specific strains (Table 1) in such groups; italic numbers inside the rectangles indicate the negative strains analyzed: 1 = *L*. *naiffi*; 2 = *L*. *shawi*; 3 = *L*. *lainsoni*; 4 = *L*. *lindenbergi*; 5 = *L*. *utingensis*.

**Figure 4 microorganisms-11-02295-f004:**
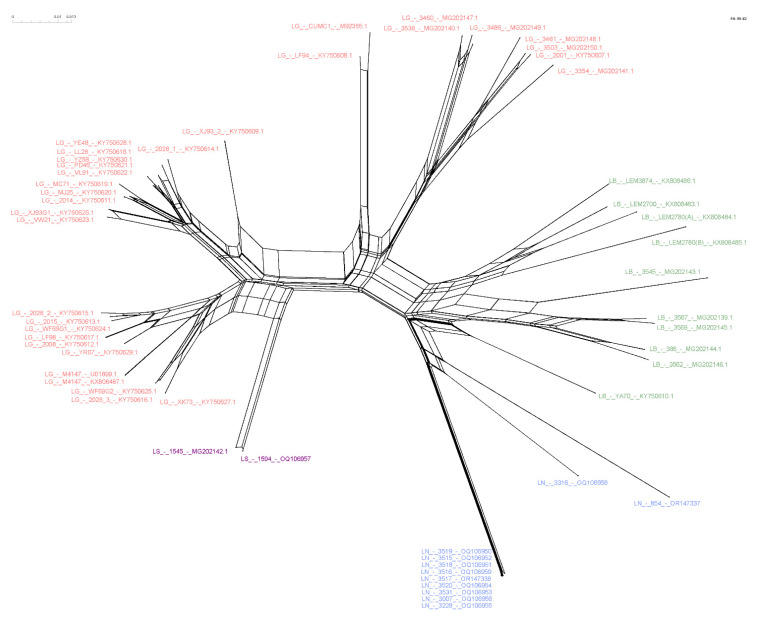
NeighborNet showing the relationship among LRV1 sequences from different *Leishmania* (*Viannia*) species. The network was computed using SplitsTree software. Text colors refer to each analyzed group. Pink = *L. guyanensis*; purple = *L. shawi*; blue = *L. naiffi*; green = *L. braziliensis*. Strains 854 and 3316, belonging to the species of *L. naiffi*, presented a divergent profile compared to the other strains of the same group.

**Table 1 microorganisms-11-02295-t001:** *Leishmania* (*Viannia*) strains from different species screened for the presence of the viral LRV1 endosymbiont and information on accession for LRV1 sequences obtained for the positive strains.

*Leishmania* Strain ID (IOCL)	Parasite Species	*Leishmania* International Code	Geographic Origin (City, State)	Sequence Length	GenBank Accession Number
854	*L. naiffi*	ISQU/BR/1985/IM2264	Cachoeira Porteira, Pará	759	OR147337
3007	*L. naiffi*	MHOM/BR/2003/IRCF	Manaus, Amazonas	701	OQ106956
3228	*L. naiffi*	MHOM/BR/2010/MS	Manaus, Amazonas	699	OQ106955
3316	*L. naiffi*	MHOM/BR/2011/58-AMS	Mojuí dos Campos, Pará	706	OQ106958
3515	*L. naiffi*	MHOM/BR/2013/49UAS	Manaus, Amazonas	709	OQ106952
3516	*L. naiffi*	MHOM/BR/2013/63DDL	Manaus, Amazonas	799	OQ106959
3517	*L. naiffi*	MHOM/BR/2013/65HCC	Manaus, Amazonas	710	OR147338
3518	*L. naiffi*	MHOM/BR/2013/66CPS	Manaus, Amazonas	679	OQ106951
3519	*L. naiffi*	MHOM/BR/2013/51FRS	Manaus, Amazonas	660	OQ106950
3520	*L. naiffi*	MHOM/BR/2013/62FJFM	Manaus, Amazonas	632	OQ106954
3531	*L. naiffi*	MHOM/BR/2013/56EGP	Manaus, Amazonas	708	OQ106953
991	*L. naiffi*	MDAS/BR/1987/IM3307	São Félix do Xingu, Pará	-	---
992	*L. naiffi*	MDAS/BR/1987/IM3280	São Félix do Xingu, Pará	-	---
993	*L. naiffi*	MDAS/BR/1987/IM3281	São Félix do Xingu, Pará	-	---
1123	*L. naiffi*	MHOM/BR/1986/IM2736	Manaus, Amazonas	-	---
1365	*L. naiffi*	MDAS/BR/1979/M5533	Almeirim, Pará	-	---
3310	*L. naiffi*	MHOM/BR/2011/S50	Santarém, Pará	-	---
3541	*L. naiffi*	MHOM/BR/2014/61AAM	Manaus, Amazonas	-	---
1594	*L. shawi*	MHOM/BR/1990/IM2842	Manaus, Amazonas	781	OQ106957
1067	*L. shawi*	IWHI/BR/1985/IM2324	Tucuruí, Pará	-	---
1068	*L. shawi*	IWHI/BR/1985/IM2326	Tucuruí, Pará	-	---
3481	*L. shawi*	MHOM/BR/2013/18	Manaus, Amazonas	-	---
1023	*L. lainsoni*	MHOM/BR/1981/M6426	Benevides, Pará	-	---
1266	*L. lainsoni*	MCUN/BR/1983/IM1721	Tucuruí, Pará	-	---
2497	*L. lainsoni*	MHOM/BR/2002/NMT-RBO 027P	Rio Branco, Acre	-	---
3398	*L. lainsoni*	MHOM/BR/2012/AP60A	Porto Velho, Rondônia	-	---
2690	*L. lindenbergi*	MHOM/BR/1966/M15733	Belém, Pará	-	---
3645	*L. lindenbergi*	MHOM/BR/2015/RO514	Porto Velho, Rondônia	-	---
3746	*L. lindenbergi*	MHOM/BR/2014/RO285	Porto Velho, Rondônia	-	---
2689	*L. utingensis*	ITUB/BR/1977/M4964	Belém, Pará	-	---

The negative strains for LRV1 are highlighted in gray. IOCL = strain codes in the *Leishmania* Collection at Fiocruz; MHOM = Mammalia, *Homo sapiens*; ISQU = Insecta, *Lutzomyia squamiventris*; MDAS = Mammalia, *Dasypus* sp.; IWHI = Insecta, *Lutzomyia whitmani*; MCUN = Mammalia, *Cuniculus* sp.; ITUB = Insecta, *Lutzomyia tuberculata.*

**Table 2 microorganisms-11-02295-t002:** Estimates of average evolutionary divergence, based on the number of differences and Tamura 3-parameter, over sequence pairs within mean groups.

	*L. guyanensis*	*L. braziliensis*	*L. naiffi*	*L. shawi*
Number of differences	33.21	34.8	13.25	1
Tamura 3-parameter model	0.133	0.139	0.053	0.003

**Table 3 microorganisms-11-02295-t003:** Estimates of evolutionary divergence over sequence pairs between groups based on the Tamura 3-parameter model (below the diagonal) and the number of differences (above the diagonal).

	*L. naiffi*	*L. shawi*	*L. guyanensis*	*L. braziliensis*
*L. naiffi*	-	42.23	43.41	42.17
*L. shawi*	0.170	-	37.99	48.05
*L. guyanensis*	0.176	0.153	-	48.32
*L. braziliensis*	0.171	0.199	0.201	-

## Data Availability

No new data were created or analyzed in this study. Data sharing is not applicable to this article.

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
