# Peer review of "Exploring Host-Specificity: Untangling the Relationship between Leishmania (Viannia) Species and Its Endosymbiont Leishmania RNA Virus 1"

_microorganisms, 2023, doi:10.3390/microorganisms11092295_

Round 1

Reviewer 1 Report

This is a very important report on the LRV1-host specificity. It must be published because the novelty of the findings is unquestionable. Kudos to authors! The new data are extremely valuable for the community. 

Now, about the problems. The manuscript is presented in a very inconsistent and sloppy way and this must be improved! I will mention just some (out of many) points that require authors attention, but the whole manuscript should be revised for clarity. 

1) Please spell last names of authors in a way that would be consistent with previous  publications.

2)  Please remove the graphical abstract. 

3) I am very surprised that a relevant paper 10.3390/v13112305 is not cited and discussed.

4) The ref. 1 is outdated. Please cite WHO 2023. 

5) One of the most important papers (Ives at al.) is cited in passing. I suggest to elaborate on it more. 

6) Ref. 11 is simply wrong! But, I agree, the structures must be mentioned: 10.1128/jvi.01957-20 and 10.1016/j.virol.2022.09.014. 

7) I seriously doubt that only 5E5 parasites were used for RNA preparation.

8) Please unify spelling of companies (I suggest: name, city, country).

9) Please revise and unify the references (word capitalization, Italics for taxa, etc.)   

10) I believe that the taxonomically-correct spelling is Leishmania RNA virus 1 (all in Italic) and this should be unified in the text and the references. 

Presentation MUST be improved language-wise for clarity. I suggest it can be sent to a professional agency or (better) a native speaker.  

Author Response

This is a very important report on the LRV1-host specificity. It must be published because the novelty of the findings is unquestionable. Kudos to authors! The new data are extremely valuable for the community. 

Now, about the problems. The manuscript is presented in a very inconsistent and sloppy way and this must be improved! I will mention just some (out of many) points that require authors attention, but the whole manuscript should be revised for clarity. 

Authors: Thank you for the nice revision of our manuscript. We accepted most of the suggestions made by this reviewer. We also submitted the manuscript for English revision by a company suggested by MDPI. Hope now we are presenting our results in a better way,  consistently, clearly  and scientifically sound.

1) Please spell last names of authors in a way that would be consistent with previous  publications.

Authors: We apologize for this, but we did not understand the point raised by the reviewer here. If he/she is mentioning the names of authors of this study, they are all correct; the only difference was Cantanhêde, that for some reason the circumflex was missed, but we included it now. If the problems were with authors from the listed references, we checked all again.

2)  Please remove the graphical abstract. 

Authors: We didn’t understand why the reviewer suggested removing the graphical abstract. We think it summarizes our study very well and we prefer to keep it if the Editors agree with our decision. We think that we prepared a nice graphical abstract that might be used by other authors in different kinds of presentations.

3) I am very surprised that a relevant paper 10.3390/v13112305 is not cited and discussed.

Authors: Yes, this is a very important study that we forgot to mention because we focused on studies related to LRV1 and Leishmania (Viannia) species. We deeply apologize for that!. In the revised version this study was added at the introduction and also as part of the discussion on host-specificity for LRV.

4) The ref. 1 is outdated. Please cite WHO 2023. 

Authors: The reference was updated as suggested.

5) One of the most important papers (Ives at al.) is cited in passing. I suggest to elaborate on it more. 

Authors: We completely agree that the study published by Ives et al is one of the most important papers showing the participation of LRV1 in the outcome of cutaneous leishmaniasis. We explore this study a little more at the introduction of this new version, but not very deeply since there are many other studies exploring that and the study by Ives et al is dealing with a subject different from our study. 

6) Ref. 11 is simply wrong! But, I agree, the structures must be mentioned: 10.1128/jvi.01957-20 and 10.1016/j.virol.2022.09.014. 

Authors: Thanks for noting that. The reference was inadequate and was removed. Considering that there are many papers already mentioning the structure of both LRV1 and LRV2 and that this is not the subject of our study, we prefer not to add information on that in our study, since this will not add any relevant information to the results that we are presenting.

7) I seriously doubt that only 5E5 parasites were used for RNA preparation.

Authors: We do not provide the final volume used for the subsequent steps, as it is such a routine stage within our protocols. We only report the cell volume per mL, which is 5.105, but we perform the extraction from 10mL of promastigotes culture. We have corrected the information in the text, considering the total cell volume, which in 10 mL is 5.106. With this quantity of parasites, the RNA extraction yield ranges between 1 – 2µg per microliter. The maximum RNA concentration (2µg) is utilized for the cDNA synthesis, to ensure the detection of LRV1, even at low concentrations.       

8) Please unify spelling of companies (I suggest: name, city, country).

Authors: Done

9) Please revise and unify the references (word capitalization, Italics for taxa, etc.)   

Authors: Done 

10) I believe that the taxonomically-correct spelling is Leishmania RNA virus 1 (all in Italic) and this should be unified in the text and the references. 

Authors: Done 

Reviewer 2 Report

  • 1.This article has certain scientific significance and potential application value for the clinical treatment of cutaneous leishmaniasis, and it is suggested to be revised and published. 2. L. lainsoni (n=4), L. lindenbergi (n=3), L. naiffi (n= 18), L. shawi (n=4) and L. utingensis (n= 1) were test , but L. naiffi,L. shawi were positive in detectiong LRV1, why? 3. To determine which strains deposited in the Fiocruz Leishmania Collection (CLIOC) 111were positive for LRV1, a screening protocol was performed using the primers LRV F - 5´- 112 ATGCCTAAGAGTTTGGATTCG- 3’ and LRV R - 5´- ACAACCAGACGATTGCTGTG – 1133’ 。The primers specifity is good not good. why ? Why is there only one primer and no control. 3.It is desirable to have the results of experimental studies with electron microscopy of the virus. 4.It would be even better if we could isolate and culture the virus.

/

Author Response

1.This article has certain scientific significance and potential application value for the clinical treatment of cutaneous leishmaniasis, and it is suggested to be revised and published.

Authors: Thank you for your revision. We hope that now our revised manuscript is adequate for publication 

  1. L. lainsoni (n=4), L. lindenbergi (n=3), L. naiffi (n= 18), L. shawi (n=4) and L. utingensis (n= 1) were test , but L. naiffi,L. shawi were positive in detectiong LRV1, why? 

Authors: This is a very interesting question, but we do not have an answer for that yet. We don't know why some parasites present LRV1 and others do not. This is also observed for parasites from the same species circulating in the same region. For example, in our study we detected LRV1 in 11 L. naiffi strains, but seven were negative for LRV1. Considering the limitation of samples tested, we cannot affirm that parasites classified as L. lainsoni, L. lindenbergi and L. utingensis are not bearing LRV1, but those evaluated in the present study are negative for this virus. We are trying to get more samples for each species to continue this investigation. Concerned about this, a very interesting aspect is related to the presence of LRV1 in L. braziliensis, since this species is widespread in Central and South America, but the presence of parasites bearing LRV1 is quite restricted to parasites circulating in the Amazon Region. We have some ongoing projects aiming to understand LRV1-L. (Viannia) species relationship and the possibilities of co-evolution.   

  1. To determine which strains deposited in the Fiocruz Leishmania Collection (CLIOC) were positive for LRV1, a screening protocol was performed using the primers LRV F - 5´- ATGCCTAAGAGTTTGGATTCG- 3’ and LRV R - 5´- ACAACCAGACGATTGCTGTG –3’ 。The primers specifity is good not good. why ? Why is there only one primer and no control. 

Authors:  We added some information about controls in this new version: we performed PCR targeting Leishmania hsp70 gene as endogenous control, to check the quality of cDNA. Furthermore, a well-known LRV1 positive strain (M4147 - L. guyanensis) was used as positive control for all sets of PCR conducted.

  1. It is desirable to have the results of experimental studies with electron microscopy of the virus. 

Authors: There are some already published studies showing Leishmania virus by electron microscopy (10.1073/pnas.85.24.9572; 10.1128/JVI.01957-20). We don't think that results obtained by electron microscopy will add any information to our study, considering the hypothesis we pursued. The main goal of our study was to verify the presence of LRV1 in strains of different L. (Viannia) species and conduct phylogenetic analysis to see the relationship among LRV1 sequences from different L. (Viannia) species, assuming as hypothesis that there is hos-specific relationship between LRV1 and L. (Viannia) species

  1. It would be even better if we could isolate and culture the virus.

Authors: This is an interesting point, but we think that is more appropriate for other kinds of studies. Some researchers already tried to isolate and cultivate Leishmania RNA Virus, but so far it is only possible to maintain LRV inside Leishmania parasites. It is possible to cure parasites from LRV1 using different approaches, including by treating the parasites with adenosine analogs, such as 2′C-methyladenosine. It is also possible to transfer LRV1 to a LRV1-free parasite by extracellular vesicles. Now, in our lab we are trying to transfer LRV1 from one species to another and then we will check the interference of LRV1 in their non-host parasites. We think that this is a very important issue, but not essential to the manuscript we are presenting now.  

Round 2

Reviewer 1 Report

All my concerns were addressed. 

Reviewer 2 Report

/